# Analysis of Antioxidant Constituents of Filtering Infusions from Oak (*Quercus sideroxyla* Bonpl. and *Quercus eduardii* Trel.) and Yerbaniz (*Tagetes lucida* (Sweet) Voss) as Monoamine Oxidase Inhibitors

**DOI:** 10.3390/molecules28135167

**Published:** 2023-07-02

**Authors:** Saúl Alberto Álvarez, Nuria Elizabeth Rocha-Guzmán, Jorge Alberto Sánchez-Burgos, José Alberto Gallegos-Infante, Martha Rocío Moreno-Jiménez, Rubén Francisco González-Laredo, Santiago Solís-González

**Affiliations:** 1Research Group on Functional Foods and Nutraceuticals, Department of Chemical and Biochemical Engineering, TecNM/Instituto Tecnológico de Durango, Felipe Pescador 1830 Ote., Durango 34080, Dgo., Mexico; 10040546@itdurango.edu.mx (S.A.Á.); agallegos@itdurango.edu.mx (J.A.G.-I.); mrmoreno@itdurango.edu.mx (M.R.M.-J.); rubenfgl@itdurango.edu.mx (R.F.G.-L.); 2Postgraduate Program in Food Sciences, TecNM/Instituto Tecnológico de Tepic, Avenida Tecnológico, Número 2595, Colonia Lagos del Country, Tepic 63175, Nayarit, Mexico; jsanchezb@tepic.tecnm.mx; 3TecNM/I.T. El Salto, Calle Tecnológico # 101, Col. La Forestal, El Salto 34942, P.N. Durango, Mexico; sugisolis@yahoo.com.mx

**Keywords:** anxiety, depression, in silico, phenolic compounds

## Abstract

The antioxidant constituents of ancestral products with ethnobotanical backgrounds are candidates for the study of filtering infusions to aid in pharmacotherapies focused on the treatment of depression and anxiety. Monoamine oxidase A (MAO-A) is an enzyme that regulates the metabolic breakdown of serotonin and noradrenaline in the nervous system. The goal of this study was to evaluate in vitro and in silico the effect of antioxidant constituents of filtering infusions from yerbaniz (*Tagetes lucida* (Sweet) Voss) and oak (*Quercus sideroxyla* Bonpl. and *Quercus eduardii* Trel.) as monoamine oxidase inhibitors. Materials were dried, ground, and mixed according to a simplex–centroid mixture design for obtaining infusions. Differential analysis of the phenolic constituent’s ratio in the different infusions indicates that among the main compounds contributing to MAO-A inhibition are the gallic, chlorogenic, quinic, and shikimic acids, quercetin glucuronide and some glycosylated derivatives of ellagic acid and ellagic acid methyl ether. Infusions of *Q. sideroxyla* Bonpl. leaves, because of their content (99.45 ± 5.17 µg/mg) and synergy between these constituents for MAO-A inhibition (52.82 ± 3.20%), have the potential to treat depression and anxiety. Therefore, future studies with pharmacological approaches are needed to validate them as therapeutic agents with applications in mental health care.

## 1. Introduction

Depression and anxiety are problems that have affected all health systems worldwide. From the early stages of these alterations, the quality of life of individuals is affected by the fact that they can simply become a chronic disorder. In the pre-pandemic era, the World Health Organization estimated that one in six working-age adults suffered from some type of mental disorder. During the pandemic period, depression and anxiety increased by more than 25%, impacting the global economy to the tune of approximately USD 1 trillion per year [1]. This scenario has led the population to develop greater awareness, demanding from the market the supply of foods and beverages that help mitigate these mental disorders. Particularly during the COVID-19 era, an increase in demand has been observed in the supply of natural sources aimed at the prevention and management of depression and anxiety. Based on this background, companies have considered ancestral sources consumed for food and medicinal purposes as a basis for the discovery of new drugs and for the innovation of these ancestral matrices to offer new products to consumers [2].

There is currently a wide variety of drugs on the market with different mechanisms of action for the prevention and treatment of depression, which are grouped into three main categories: monoamine oxidase inhibitors (MAOIs), tricyclic antidepressants (TCAs), and second-generation antidepressants. The latter include norepinephrine reuptake inhibitors (NRIs), selective serotonin reuptake inhibitors (SSRIs), and serotonin-norepinephrine reuptake inhibitors (SNRIs) [3]. However, not all treatments have the expected results; even clinical evidence has shown the appearance of important adverse effects that negatively affect the health status of patients. Thus, it has become necessary to find alternative therapies with similar efficacy, seeking to reduce these effects.

Monoamine oxidase A is a flavoprotein that metabolizes serotonin and norepinephrine. The deficiency of these neurotransmitters is directly related to the development of symptoms of depression and anxiety [4]. In this context, sources or components capable of inhibiting monoamine oxidase A are considered important strategies that lead research groups to return to the study of ancestral matrices of food used for these purposes. This has led to the fact that currently, from a perspective oriented to the beverage market, products that have traditionally been identified as infusions or teas are positioned as infused waters or tisanes. In this perspective, infusions made from herbal products are currently consumed as an herbal refreshment, rich in bioactive compounds, with the expectation that through their consumption, risk factors for certain diseases are reduced [5]. Improved consumer awareness is demanding the innovation of ancestral foods and beverages, raising, as specific needs, the supply of filtering infusions and infused waters derived from natural products by linking them to health and wellness.

In silico studies are used to explore those molecular docking analyses widely used in enzymology, as well as for computer-assisted drug design [6]. These analyses are becoming increasingly relevant in drug research and development since they allow testing at the molecular level to find substances with greater medicinal potential and less toxicity. These studies stand out for their versatility and precision in predicting the physicochemical properties of organic binders. One of the most widely used tools is based on the search for the optimal or most favorable conformation and position of a ligand within a molecular target or two macromolecules [7].

In summary, herbal matrices have generated great interest in researchers for their medicinal effects and the benefits they can provide to health. This has led to the exploration of new, more effective, accessible, and safe treatments for the population. This research contemplates the study of filtering infusions with yerbaniz (*Tagetes lucida* (Sweet) Voss), oak leaves (*Quercus sideroxyla* Bonpl. and *Q. eduardii* Trel.), and their mixtures, chosen based on their composition of hydroxycinnamic acids and flavonols, chemical groups recently related to anxiolytic and antidepressant effects.

## 2. Results and Discussion

### 2.1. Physicochemical Characterization

The °Brix, pH, and total titulable acidity values of the ten samples were different. It was clear that the combination of these herbal sources promoted changes in the values of these physicochemical parameters, which may have been related to different combinations of the matrices under study and their chemical constituents. To specify an important physicochemical parameter as the °Brix values, which represents the number of solids or total dry matter dissolved in a given liquid, all other samples showed values below 0.5%. The pH values ranged from 5.13 to 6.39 for the mixtures, while for the *Q. sideroxyla* Bonpl. infusions, the values were 5.04 ± 0.06, while for *Q. eduardii* Trel. They were 5.45 ± 0.02. Gamboa-Gómez et al. [8] indicate similar values for oak infusions (*Q. convallata* and *Q. arizonica*). The total titratable acidity values for the different samples ranged from 0.01 to 0.06 g of citric acid/L (Figure 1). These responses may be caused by the differences in phenolic profiling present in infusions (Table 1), sugars, and other aqueous extractables.

### 2.2. Chemical Constituents

Twenty phenolic acids, eight flavonoids, and eleven hydrolyzable tannins were identified in filtering infusions by UPLC-PDA-ESI^-^-MS/MS (Table 1). The highest diversity and phenolic content were identified in the FI obtained with *Q. sideroxyla* Bonpl. (99.45 ± 5.17 µg/mg), finding a clear example of dilution when mixed with the other two sources under study. The phenolic acid ratios with respect to the total content identified and quantified represented 54.01% of phenolic acids, 10.07% of flavonoids, and 35.91% of hydrolyzable tannins for this FI (Table 2).

The FI of *Q. eduardii* Trel. had 51.77% less content of phenolic compounds (47.96 ± 1.34 µg/mg), with respect to *Q. sideroxyla* Bonpl., with 68.8% of phenolic acids, 9.33% of flavonoids, and 21.79% of hydrolyzable tannins. Similar contents have been reported by other authors with respect to the content of these chemical groups for the oak species under study [9]. In oak FI, their content of quinic, shikimic, and chlorogenic acids, quercetin glycosides, kaempferol glucoside, as well as ellagitannins stands out (Table 3). 

Some of these constituents are widely documented for their antioxidant effect [10]. For its part, the FI with the lowest phenolic content was *Tagetes lucida* (Sweet) Voss (15.12 ± 0.23), with 65.16% of phenolic acids highlighting in its content of quinic, chlorogenic, caffeic, and protocatechuic acids (Table 3).

### 2.3. Antioxidant Activity

Polyphenols are organic compounds found in a balanced diet and are commonly consumed in herbal teas. Polyphenolic compounds extracted with high polarity solvents such as water exhibit outstanding antioxidant activity [11]. As mentioned above, the phenolic profiling identified indicates the presence of constituents in the FI that are recognized for their ability to inhibit reactive oxygen species, reducing power [12], and free radical scavenging [13]. 

Our results indicate that the combination of *Q. sideroxyla* Bonpl. infusion constituents can absorb oxygen radicals, have higher reducing power, and are able to outstandingly scavenge free radicals (Table 4). In general, the compounds that match all the antioxidant responses identified through Pearson’s correlation analysis with R^2^ values > 0.60 were the gallic acid, procyanidin B1, catechin, quercetin glucoside, kaempferol glucoside, ellagic acid hexoside, digalloyl-HHDP-hexoside, and pentagalloyl glucose. In addition, quercetin glucuronide and quinic, shikimic, and chlorogenic acids showed a significant influence in absorbing oxygen radicals and can scavenge free radicals, whereas vescavaloninic acid and 2-hydroxybenzoic acid can absorb oxygen radicals.

In summary, the fact of having matrices with the capacity to respond to different antioxidant mechanisms leads us to hypothesize that sources such as *Q. sideroxyla* Bonpl. may be a candidate in the prevention of mental disorders since events such as neuronal loss are related to elevated levels of oxidative stress [2].

### 2.4. Inhibition of MAO-A In Vitro and In Silico

It is increasingly common to find reports demonstrating that constituents of herbal infusions have MAO-A inhibitory effects. This supports the rescue of knowledge promoting mental health from unconventional sources that were ancestrally consumed for the detection of bioactive natural products with MAO-A inhibitory activity. MAO-A has been shown to have a high selectivity for the neurotransmitter serotonin, so inhibitors of this enzyme can prolong the action of this neurotransmitter and are useful for the treatment and prevention of disorders such as anxiety and depression [14].

Figure 2a shows the response associated with MAO-A inhibition. Again, the best response was obtained for *Q. sideroxyla* Bonpl. infusion, followed by *T. lucida* (Sweet) Voss infusion and their combination in a 1:1 ratio. It is important to note that the infusion of *Q. eduardii* Trel. did not show an inhibitory effect on MAO-A activity, as well as the combinations in which *T. lucida* (Sweet) Voss was mainly present in the formulation. In our study, the enrichment of the filtering infusions with *Tagetes lucida* (Sweet) Voss was chosen from a previous study [15]. It refers to the consumption of these infusions with anxiolytic and sedative-like effects by involving serotonergic and GABAergic neurotransmission associated with coumarinic constituents such as dimethylfraxetin and flavonoids of the flavonol group such as quercetin 3-O-glucoside and rutin. These compounds represented 16.17 and 11.98% of the total flavonoid content identified and quantified in this study for the FI of yerbaniz. It is important to point out that the studies conducted by Pérez-Ortega et al. [15] contemplated aqueous extracts obtained at 21.88% from the foliar part of the plant, a concentration an order of magnitude higher than those used as a maximum limit in our study (1%).

In particular, the inhibitory responses towards MAO-A by the synergy of the constituents of the infusion of *Q. sideroxyla* Bonpl. were associated with its flavonol composition since several studies indicate that kaempferol, quercetin, and their glycosides can inhibit this enzyme [10,16,17]. However, Pearson’s correlation analysis did not show significant correlations with any of the constituents of the mixtures. Thus, it was decided to perform a partial least-squares-discriminant analysis (PLS-DA), determining that the score plot (Figure 2b) evidences a clear separation between samples. A cumulative variance of 82.4% is pointed out, with component 1 explaining 64.1% of the variance. No outliers outside the confidence thresholds were observed in the analysis. In addition, the cross-validation results showed a Q^2^ value = 0.582 and R^2^ = 0.776 for the model. These scores indicate that the model data are reliable in describing the responses and that the different sets show statistical significance. So, in this analysis, the important features of the discriminant model were defined as those phenolic compounds whose VIP value was greater than 1. In this way, five phenolic acids (quinic, chlorogenic, caffeic, shikimic, and gallic acids), three flavonoids (quercetin glucuronide, epicatechin, and rutin), and four hydrolyzable tannins (ellagic acid xyloside, ellagic acid hexoside, ellagic acid methyl ether, and ellagic acid rhamnoside) (VIP > 1) showed the greatest influence on the MAO-A inhibition (Figure 2c). In summary, screening of natural sources allowed us to identify *Quercus sideroxyla* Bonpl. as a very abundant source of phenolic constituents whose synergy inhibits the activity of MAO-A, an enzyme that regulates the metabolic degradation of serotonin and noradrenaline in the nervous system.

Molecular docking was performed for compounds with VIP > 1, although for some of them, there are already reports in the literature of the binding energy and in vitro concentrations needed to reach a CI_50_ [18,19,20]. The results of molecular docking were visualized in USCF Chimera and evaluated using the FullFitness (FF) parameter (Figure 3). In general, the lower values of the FullFitness score mean that the molecule (ligand) is more likely to bind to the protein target with high affinity and specificity [21]. The results obtained showed that caffeic acid had the lowest FullFitness score (−3224.09 Kcal/mol), lower than that of the selective and non-selective inhibitors. This phenolic acid was detected in higher concentrations in *T. lucida* (Sweet) Voss (0.75 µg/mg) and in those mixtures where this herbal source was present. Caffeic acid has been studied in silico by Carpéné et al. [22], demonstrating that it is a good MAO inhibitor; however, it should not be overlooked that this compound, in general terms, is found in a low concentration with respect to other phenolics such as gallic acid. This compound showed the second-best FullFitness score (−3207.74 Kcal/mol) and was found in higher abundance in *Q. sideroxyla* Bonpl. (0.93 µg/mg). Bhuia et al. [23] indicate that gallic acid has an anxiolytic and antidepressant potential.

Finally, chlorogenic acid, shikimic acid, epicatechin, and quinic acid, despite having a lower FullFitness score than clorgiline, were more efficient than the non-selective inhibitors included in the analysis. Quinic acid and shikimic acid were the phenolic acids identified with the highest abundance in *Q. sideroxyla* Bonpl. with concentrations of 29.91 and 13.57 µg/mg, respectively. Several authors have reported that quinic acid has MAO inhibitory activity [24,25]. Similarly, shikimic acid has shown inhibitory activity in this enzyme [26]. In our study and as mentioned, the sample with the highest abundance of these compounds was *Q. sideroxyla* Bonpl., which showed the highest inhibitory effect on MAO-A in the in vitro assay (Figure 2a).

The presence of compounds in all experimental mixtures with high negative FullFitness scores, higher or close to those shown by selective or non-selective inhibitors, supports the potential use of filtering infusions as antidepressant and anxiolytic sources. However, the influence of other bioactive compounds and their synergistic effects need to be further investigated. 

In summary, clear relationships were observed between the number of phenolic compounds present in the experimental samples and the FullFitness score for individual compounds. This behavior could be related to additive or synergistic effects in the experimental mixtures. Also, they may suggest the presence of other bioactive molecules able to inhibit MAO-A. This behavior has been reported for docking experiments and plant extracts by other authors [27].

## 3. Material and Methods

### 3.1. Reagents

Phenolic acids standards (quinic, shikimic, gallic, ellagic, protocatechuic, chlorogenic, vanillic, caffeic, syringic, coumaric, ferulic, benzoic, trans-cinnamic, 3,4-di-caffeoylquinic, and rosmarinic), flavonoids standards (catechin, epi-catechin, procyanidin B1, quercetin 3-O-ß-glucuronide, quercetin 3-O-glucoside, kaempferol 3-O-glucoside, and taxifolin), and 2,4,6-trihydroxybenzaldehyde, were obtained from Sigma–Aldrich (St. Louis, MO). All reagents used for standard preparation were methanol LC-MS grade. Also, fluoresceine, 6-hydroxy-2,5,7,8-tetramethylchroman-2-carboxylic acid (Trolox), 2,2′-azobis (2-amidinopropano) dihydrochloride (AAPH), 2,4,6-tris(2-pyridyl)-s-triazine (TPTZ), ammonium persulfate, 2,2′-azino-bis(3-ethylbenzothiazoline-6-sulfonic acid) diammonium salt (ABTS), 2,2-diphenyl-1-picrylhydrazyl (DPPH), clorgiline, tranylcypromine, and MAO-A enzyme were acquired from Sigma–Aldrich (St. Louis, MO, USA).

### 3.2. Raw Material Collection and Processing 

The plant material was completely identified by botanist Socorro Gonzalez-Elizondo from the herbarium of the CIIDIR-IPN campus in Durango. Yerbaniz (*Tagetes lucida* (Sweet) Voss) with voucher number 61,483 was collected in Pueblo Nuevo, Durango (Longitude: 105°21′43″ E Latitude: 23°46′46″ N), *Quercus sideroxyla* Bonpl. leaves with voucher number 61,484, and *Q. eduardii* Trel. leaves with voucher number 61,485 were collected in Durango Parque “El Tecuán” (Longitude: 105°3’0” W, Latitude: 23°57’0” N) from July to August 2021. The collected plant material was sanitized with 1% sodium hypochlorite for 5 min. Subsequently, the material was dried at room temperature 25 °C ± 2 °C in the dark and ground to a particle size of 150 µm and finally lyophilized and stored until use.

### 3.3. Mixture Design for Extraction

A simplex–centroid mixture design was carried out for quadratic models (m = 2) (Figure 4), obtaining 1.0% by summing the different components (*Tagetes lucida* (Sweet) Voss, *Quercus sideroxyla* Bonpl., and *Quercus eduardii* Trel.) [28].

### 3.4. Preparation of Filtering Infusions (FI)

For the development of the FI, different mixtures of plant material (2.4 g) were packed in disposable tea filtering membranes (7 × 9 cm). The infusions were prepared by immersing the filter membranes in water at 80 °C at a ratio of 1:100 for 10 min. The FI were lyophilized and stored for chemical characterization and determination of nutraceutical potential (i.e., MAO-A inhibition).

### 3.5. Determination of Physicochemical Parameters

The pH values were obtained with a Sen Ion 1 Hach potentiometer according to Mexican standards [29]. The determination of total titulable acidity was carried out at room temperature, using 0.1 N sodium hydroxide, diluting a 5 mL sample of each infusion in 35 mL of distilled water, and using phenolphthalein alcohol solution at 1% concentration as an indicator. Soluble solids were determined using an ATAGO N-1E hand refractometer (Brix 0~32%) Japan and were expressed in °Brix.

### 3.6. Analysis of Phenolic Profiles by UPLC-PDA-ESI^-^-QqQ

HPLC-grade standard solutions of flavonoids and simple phenolic acids were prepared in MS-grade methanol. The samples were prepared at a ratio of 10 mg/mL (*w*/*v*), homogenized, and filtered through an acrodisk (PTFE 0.45 µm, 13 mm θ) on an insert placed in a vial with a pre-baked septum. Detection and quantification of major compounds were achieved according to Díaz-Rivas et al. [30] using electrospray ionization/tandem spectrometry in multiple reaction monitoring (MRM) mode. Briefly, sample analysis was carried out with an Acquity UPLC system (Waters Corp., Milford, MA, USA) coupled with a tandem Xevo TQ-S triple quadrupole mass spectrometer (Waters Corp., Milford, MA, USA). The LC system consisted of a sample manager (6 °C) and a quaternary solvent manager. The column used to separate the phenolic compounds was a reversed-phase Acquity^®^ BEH C18 column (1.7 µm particle size, 50 mm × 2.1 mm ID) operated at 35 °C. 

The elution gradient started with 3% B at 1.23 min, and a gradient up to 9% B was applied; at 3.82 min, the gradient increased to 16% B; at 11.40 min, the gradient reached 50% B; at 13.24 min, the gradient returned to 3% B and was maintained until 15 min to stabilize the column at a flow rate of 250 µL/min. Multiple reactions ionization mode was used for MS/MS assays. Electrospray ionization (ESI) in negative conditions were as follows: capillary voltage 2.5 kV, desolvation temperature 400 °C, source temperature 150 °C, desolvation gas flow 800 L/h and cone gas flow 150 L/h, collision gas flow 0.13 mL/min, MS mode collision energy 5.0, and MS/MS mode collision energy 20.0. For the identification of the phenolic profile (phenolic acids and flavonoids), standards (20 ng/µL) were used for monitoring retention time and *m/z* values and MS/MS transitions. A multiple-reaction monitoring mode was recorded for samples and standards. For the identification of hydrolyzable tannins, the chemical characterization was performed according to the fragmentation patterns reported by García-Villalba et al. [9] and its maximum lambda at 360 nm. The UPLC and Tandem Xevo TQ-S triple quadrupole mass spectrometer control and data processing used MassLinx v. 4.1 Software (Waters Corp).

### 3.7. Antioxidant Characterization

#### 3.7.1. Oxygen Radical Absorption Capacity (ORAC)

The oxygen radical absorbance capacity (ORAC) was performed on fermented beverages as described by Ou et al. [31]. Briefly, in a 96-well dark plate, 20 µL of the sample was reacted with 200 µL of fluorescein at 1.09 µM for 15 min at 37 °C. A first reading was recorded at 485 nm excitation and 580 nm emission wavelengths on a Synergy HT Multi-Detection Microplate Reader (Bio-Tek, Winooski, VT, USA). Later, 75 µL of 2,2-Azobis(2-methylpropionamidine) dihydrochloride (AAPH) was added to the reaction, and kinetics were started by recording the reading every 3 min for 2 h under the described conditions. Finally, the area under the curve was calculated using SigmaPlot R version 14.0 software (Systat Software Inc., 2014). PBS was used as a blank, Trolox was used as standard, and the results were expressed in µM trolox equivalents per mL of beverage.

#### 3.7.2. Ferric-Reducing Antioxidant Power (FRAP)

The ferric-reducing power was established according to Benzie and Strain [32]. Briefly, FRAP solution was prepared with an acetate buffer at 400 mM and pH 3.6, Tris (2-pyridyl)-s-triazine (TPTZ) solution at 30 mM in 40 mM HCl, and ferric chloride (FeCl_3_.6H_2_O) solution at 60 mM, in a 10:1:1 ratio. For the reaction, 20 µL of a sample or blank was used, adding 180 µL of the radical solution and incubating in the dark for 10 min before recording the absorbance at 593 nm using a Synergy HT Multi-Detection Microplate Reader (Bio-Tek, Winooski, VT, USA). Trolox was used as standard, and the reducing power was expressed as µM Trolox equivalents.

#### 3.7.3. ABTS Cation Radical Scavenging

The antioxidant capacity was evaluated using the ABTS radical scavenging method, which uses the 2,2’-azinobis (3-ethyl-benzthiazoline-6-sulfonic acid) radical described by Re et al. [33]. Briefly, the stock solution of the ABTS radical for the assay was prepared at a concentration of 7 mM in a 1:1 ratio with 2.5 mM ammonium persulfate and incubated at room temperature (25 °C) in the dark for 16 h. For the working solution, the stock solution was diluted in phosphate buffer (50 mM, pH 7.4) to an absorbance of 0.7 at 750 nm. For the reaction, 10 µL of the sample was mixed with 190 µL of a diluted ABTS solution while shaking the plate, then allowed to stand for 10 min, and absorbance was recorded at 750 nm using a Synergy HT Multi-Detection Microplate Reader (Bio-Tek, Winooski, VT, USA). Trolox was used as a standard, and data were expressed as µg Trolox equivalents per mL.

#### 3.7.4. Chain-Breaking Activity

The chain-breaking activity was determined as described by Manzocco et al. [34] with some modifications. The reaction was initiated by the addition of 30 µL of samples in 970 µL of DPPH solution in 6.1 × 10^−5^ M methanol. DPPH bleaching was followed at 515 nm (Jenway 6705 UV/Visible Scanning Spectrophotometer, Stone, Staffs, UK) at 25 °C for 5 min, recording data every 30 s. In all cases, the rate of DPPH bleaching was proportional to the concentration of the sample added to the medium. The following equation was used to obtain the reaction rate, k:(1)$$\frac{1}{A^{3}}-\frac{1}{A_{0}^{3}}=-3kt$$
where *A*_0_ is the initial optical density, and *A* is the optical density at increasing time, *t*. The chain-breaking activity was expressed as k/mg_d_._m_ (−O.D.^−3^/min/mg_d_._m_).

### 3.8. MAO-A Assay

The assay was performed in 96-well microtiter plates using the monoamine oxidase A (MAO-A) activity detection kit (fluorometric) Sigma–Aldrich MAK295. For the assay, tyramine was used as a nonspecific substrate for MAO-A, and clorgiline as a specific and irreversible inhibitor. In the reaction mixture in each well 10 μL of the extract in DMSO (1 mg/mL), or DMSO as blank, 10 μL MAO-A assay buffer, and 50 μL of the enzyme solution was added. Also, a control was performed where the sample was replaced by 10 μL of the selective inhibitor and a blank for each sample where the enzyme-working solution was replaced by 50 μL of 10 mM H_2_O_2_. Subsequently, samples were incubated for 10 min at 25 °C, and 40 μL of the substrate solution was added to each reaction well using a plate shaker to mix and finally to measure fluorescence kinetically at a wavelength for excitation at 537 nm and for emission at 587 nm at 25 °C from 10 to 30 min on a Synergy HT Multidetector Microplate Reader (BioTek, Winooski, VT, USA).

### 3.9. Evaluation of MAO-A Inhibitory Activity In Silico

The molecular docking study was performed to determine the possible interactions that could occur between the phenolic constituents of the filtering infusions and the MAO-A enzyme, following the protocols proposed by Chávez-Fumagalli et al. [35] and Ortiz et al. [7]. Ligands were obtained from the PubChem chemical molecule database. Energy minimization was performed with molecular mechanics using the Hartree–Fock algorithm (UCSF Chimera 1.14 software). Once the calculation was completed, the final structure was saved in mol2 format for further manipulation. The 3D structure of MAO-A in PDB format was obtained from the Swiss Institute for Bioinformatics (SIB). Finally, after obtaining the ligands and the target protein, the docking procedure was performed and optimized by the SIB website service. The molecular docking results were visualized in USCF Chimera and evaluated by the FullFitness (FF, spontaneity of enzyme–ligand complex formation) parameter, calculated by averaging the 30% of the most favorable “*n*” energies of a cluster, used to reduce the risk of a few complexes penalizing a whole cluster. Finally, the cluster with the most favorable FF was used to determine the residues involved in complex generation.

### 3.10. Statistical Analysis

The LC-MS data were processed by using MassLynx MS Software. All results in the tables were expressed as the mean ± standard deviation (SD). Data were analyzed by one-way ANOVA analysis (*p* < 0.05). Furthermore, data analysis was done with the MetaboAnalyst platform.

## 4. Conclusions

In a holistic approach involving the rescue of health-promoting knowledge, the identification of matrices with potential effects in the treatment of mental health has advanced enormously thanks to the combination of several trials that intend to screen these sources with anxiolytic and antidepressant potential in vitro.

Filtering infusions of *Q. sideroxyla* Bonpl. for their chemical constituents promise affordability, accessibility, and efficacy comparable to that of high-cost synthetic drugs. Therefore, future efforts in the study of *Quercus sideroxyla* Bonpl. seek to focus on clinical trials to confirm traditional use.

This study provides information on filtering infusions with antidepressant and anxiolytic potential, which may be the basis of future explorations of these natural products for the development of phytopharmaceuticals aimed at treating mental disorders.

## Figures and Tables

**Figure 1 molecules-28-05167-f001:**
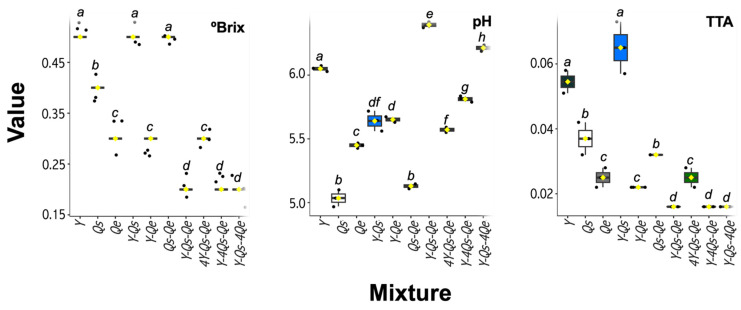
Physicochemical parameters in infusions of *Tagetes lucida* (Sweet) Voss, *Quercus sideroxyla* Bonpl., *Quercus eduardii* Trel., and their mixtures. Different letters indicate significant statistical differences between samples (Tukey, *p* < 0.05, *n* = 3).

**Figure 2 molecules-28-05167-f002:**
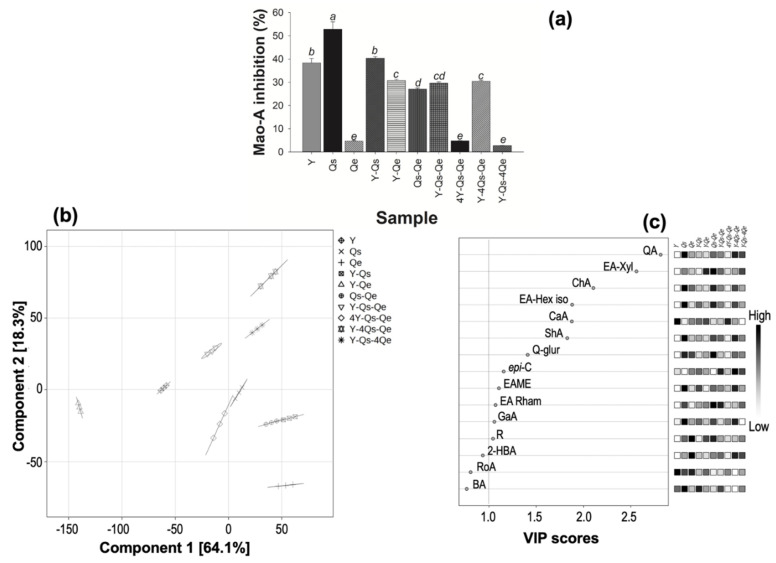
Partial least-square-discriminant (PLS–DA) of the influence of leaf content on the main phenolic profile identified by UPLC–ESI–MS/MS in mixtures. (**a**) MAO–A inhibition (%); (**b**) the results are presented as principal component score plots, and the explained variances are shown in brackets; (**c**) important features identified by PLS–DA. The color boxes on the right indicate the relative abundance of the corresponding compound in each leaf content.

**Figure 3 molecules-28-05167-f003:**
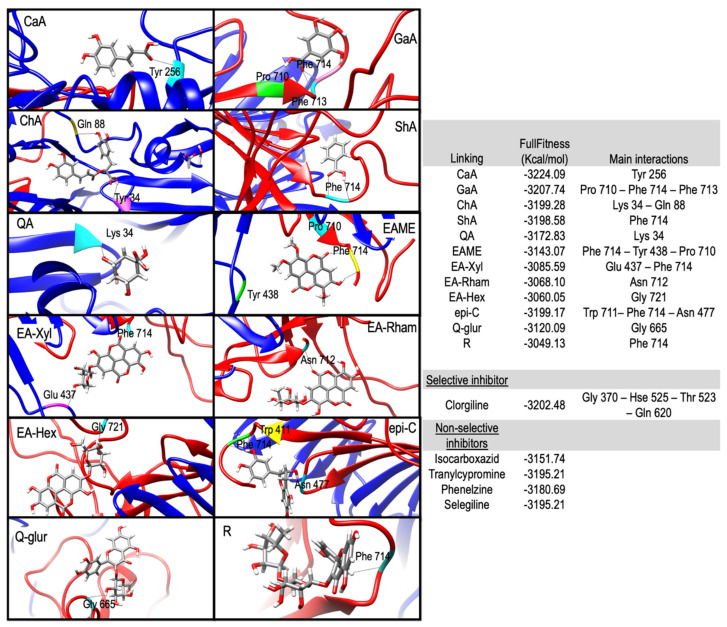
Molecular docking simulation obtained with the lowest energy conformation for important features identified by PLS–DA.

**Figure 4 molecules-28-05167-f004:**
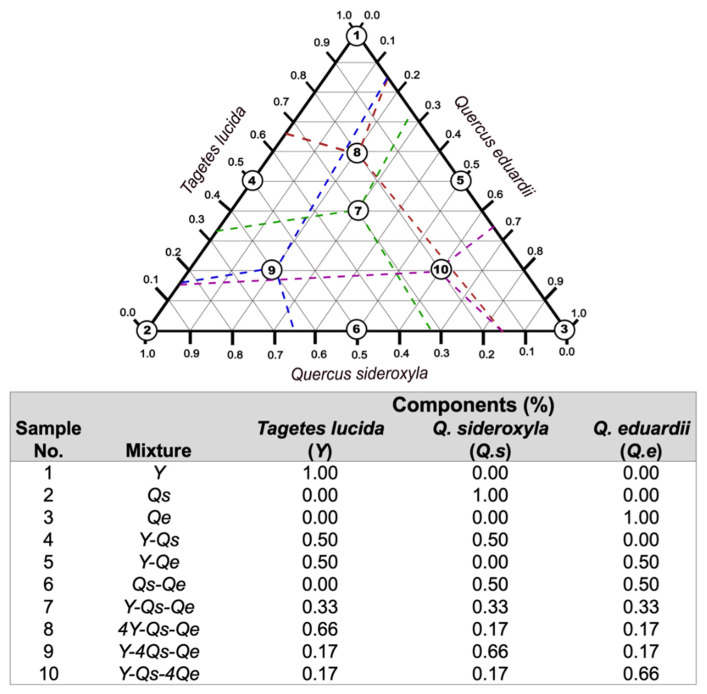
Schematic demonstration for the simplex–centroid mixture designed in percentage and acronyms assigned to each sample.

**Table 1 molecules-28-05167-t001:** Phenolic profiling identified in samples by UPLC-PDA-ESI-MS/MS.

No.	Compound	Acronym	Retention Time (Min)	λmax	Main Transitions
Phenolic acids					
1.	Quinic acid	QA	0.62	254.86	191.20 > 85.06
2.	Shikimic acid	ShA	0.68	-	173.18 > 111.07
3.	Gallic acid	GaA	1.48	270.86	169.15 > 125.05
4.	Protocatechuic acid	PA	2.74	258.86, 292.86	153.15 > 109.05
5.	Caffeoylquinic acid isomer	CQA iso	3.41	324.86	353.10 > 191.20
6.	2,5-di-hydroxybenzoic acid	2,5-DHBA	3.65	323.86	153.15 > 108.92
7.	4-hydroxybenzoic acid	4-HBA	3.89	253.86	137.04 > 93.05
8.	Chlorogenic acid	ChA	4.27	324.86	353.10 > 191.20
9.	4-O-caffeoylquinic acid	4-O-CQA	4.45	324.86	353.10 > 191.20
10.	Vanillic acid	VA	4.54	259.86, 290.86	167.18 > 152.02
11.	Caffeic acid	CaA	4.73	321.86	179.19 > 135.08
12.	Syringic acid	SyA	4.86	273.86	197.21 > 182.05
13.	2,4,6-trihydroxybenzaldehyde	2,4,6-THBA	5.76	290.86	153.15 > 83.04
14.	Coumaric acid	CouA	5.86	308.86	163.24 > 119.08
15.	Ferulic acid	FA	6.42	322.86	193.24 > 134.04
16.	Benzoic acid	BA	7.02	271.86	121.10 > 77.10
17.	Trans-cinnamic acid	t-CiA	7.07	276.86	147.17 > 103.08
18.	2-hydroxybenzoic acid	2-HBA	7.12	254.53	137.04 > 93.05
19.	Di-caffeoylquinic acid	di-CQA	7.19	324.86	515.43 > 353.20
20.	Rosmarinic acid	RoA	7.55	327.86	359.28 > 161.04
Flavonoids					
21.	Procyanidin B1	PB1	3.88	278.86	577.44 > 289.18
22.	Catechin	C	4.29	277.86	288.97 > 245.06
23.	(*epi*)-catechin	*epi*-C	5.18	277.86	288.97 > 245.06
24.	Rutin	R	6.46	354.86	609.04 > 270.94
25.	Quercetin 3-O-ß-glucuronide	Q-glur	6.61	287.86	476.92 > 300.99
26.	Quercetin 3-O-glucoside	Q-glu	6.65	354.86	463.36 > 300.42
27.	Taxifolin	Ta	6.66	284.86	303.03 > 285.00
28.	Kaempferol 3-O-glucoside	K-glu	7.15	357.86	447.30 > 284.24
Hydrolyzable tannins					
29.	Di-galloyl-HHDP-hexoside	DG-HHDP-Hex	5.05	359.86	785.00 > 301.00
30.	Trigalloyl hexoside	TG-Hex	5.26	359.86	635.00 > 465.00
31.	Ellagic acid hexoside	EA-Hex	5.44	353.86	463.00 > 300.00
32.	Pentagalloyl glucose	PG-Glu	5.86	-	939.00 > 169.00
33.	Ellagic acid xyloside	EA-Xyl	6.01	359.86	433.00 > 301.00
34.	Ellagic acid rhamnoside	EA-Rham	6.18	359.86	447.00 > 300.00
35.	Ellagic acid methyl ether	EAME	6.59	359.86	315.00 > 300.00
36.	Ellagic acid hexoside isomer	EA-Hex iso	6.61	350.86	463.00 > 300.00
37.	Vescavaloninic acid	VeA	6.83	-	1101.00 > 569.00
38.	Ellagic acid rhamnoside isomer	EA-Rham iso	7.17	359.86	447.00 > 300.00
39.	Ellagic acid dimethyl ether	EADME	10.33	-	329.00 > 229.00

**Table 2 molecules-28-05167-t002:** Total phenolic compounds (µg/mg) in FI, calculated in MRM mode using UPLC-ESI^-^-MS/MS.

Mixture	Phenolic Acids	Flavonoids	Hydrolyzable Tannins	Total
*Y*	9.855 ± 0.069 ^g^	1.669 ± 0.005 ^i^	3.569 ± 0.301 ^f^	15.122 ± 0.236 ^g^
*Qs*	53.723 ± 3.784 ^a^	10.020 ± 0.281 ^a^	35.716 ± 1.104 ^a^	99.459 ± 5.170 ^a^
*Qe*	33.029 ± 0.908 ^d^	4.479 ± 0.018 ^d^	10.455 ± 2.234 ^e^	47.963 ± 1.344 ^d^
*Y-Qs*	31.418 ± 0.968 ^d^	5.012 ± 0.147 ^c^	23.848 ± 1.185 ^c^	60.280 ± 0.069 ^c^
*Y-Qe*	20.800 ± 0.804 ^e^	3.228 ± 0.075 ^g^	17.360 ± 0.826 ^d^	41.389 ± 1.704 ^e^
*Qs-Qe*	41.823 ± 1.814 ^b^	7.613 ± 0.293 ^b^	31.557 ± 1.502 ^b^	80.994 ± 3.610 ^b^
*Y-Qs-Qe*	35.630 ± 1.539 ^c,d^	4.049 ± 0.114 ^e^	18.819 ± 0.896 ^d^	58.500 ± 2.549 ^c^
*4Y-Qs-Qe*	17.156 ± 0.668 ^f^	2.652 ± 0.047 ^h^	11.029 ± 0.525 ^e^	30.838 ± 1.241 ^f^
*Y-4Qs-Qe*	47.405 ± 2.117 ^a^	7.324 ± 0.270 ^b^	25.396 ± 1.209 ^c^	80.126 ± 3.597 ^b^
*Y-Qs-4Qe*	38.843 ± 3.347 ^b,c^	3.666 ± 0.096 ^f^	16.162 ± 0.726 ^d^	58.672 ± 3.839 ^c^

Results are presented as mean ± standard deviation for (Tukey, *p* < 0.05; *n = 3*); different letters in each column indicate significant statistical differences.

**Table 3 molecules-28-05167-t003:** Phenolic compounds (µg/mg) present in samples calculated in MRM mode using UPLC-PDA-ESI-MS/MS.

Compound	Y	Qs	Qe	Y-Qs	Y-Qe	Qs-Qe	Y-Qs-Qe	4Y-Qs-Qe	Y-4Qs-Qe	Y-Qs-4Qe
Phenolic acids										
QA	2.55 ± 0.08	29.91 ± 2.82	17.03 ± 0.65	15.17 ± 0.93	9.29 ± 0.43	20.22 ± 0.95	17.58 ± 0.82	6.83 ± 0.31	25.24 ± 1.19	21.32 ± 2.01
ShA	0.67 ± 0.06	13.57 ± 0.61	7.05 ± 0.02	7.95 ± 0.45	4.21 ± 0.21	11.49 ± 0.57	9.57 ± 0.45	3.82 ± 0.18	12.32 ± 0.58	9.38 ± 0.89
GaA	0.03 ± 0.00	0.93 ± 0.10	0.20 ± 0.01	0.49 ± 0.03	0.16 ± 0.00	0.59 ± 0.03	0.04 ± 0.00	0.24 ± 0.01	0.72 ± 0.03	0.01 ± 0.00
PA	1.39 ± 0.06	0.11 ± 0.01	0.13 ± 0.03	0.41 ± 0.03	0.32 ± 0.08	0.13 ± 0.01	0.53 ± 0.02	0.41 ± 0.02	0.26 ± 0.01	0.49 ± 0.02
CQA iso	0.25 ± 0.00	0.24 ± 0.01	0.31 ± 0.00	0.26 ± 0.01	0.29 ± 0.00	0.29 ± 0.00	0.27 ± 0.00	0.25 ± 0.00	0.27 ± 0.00	0.29 ± 0.00
2,5-DHBA	traces	0.01 ± 0.00	0.01 ± 0.00	traces	traces	0.01 ± 0.00	traces	traces	traces	0.01 ± 0.00
4-HBA	0.36 ± 0.04	0.20 ± 0.00	0.19 ± 0.00	0.26 ± 0.00	0.26 ± 0.00	0.19 ± 0.00	0.24 ± 0.00	0.28 ± 0.00	0.21 ± 0.00	0.23 ± 0.00
ChA	1.03 ± 0.06	6.52 ± 0.27	5.11 ± 0.23	3.78 ± 0.41	3.37 ± 0.15	6.07 ± 0.28	4.40 ± 0.20	2.21 ± 0.09	5.84 ± 0.27	4.14 ± 0.37
4-O-CQA	0.20 ± 0.00	0.00 ± 0.00	0.20 ± 0.00	0.21 ± 0.00	0.00 ± 0.00	0.21 ± 0.00	0.22 ± 0.00	0.00 ± 0.00	0.00 ± 0.00	0.22 ± 0.00
VA	0.36 ± 0.00	0.31 ± 0.00	0.31 ± 0.00	0.33 ± 0.00	0.32 ± 0.00	0.30 ± 0.00	0.33 ± 0.00	0.34 ± 0.00	0.31 ± 0.00	0.32 ± 0.00
CaA	0.75 ± 0.02	0.28 ± 0.01	0.35 ± 0.00	0.36 ± 0.01	0.46 ± 0.01	0.28 ± 0.00	0.30 ± 0.00	0.63 ± 0.02	0.32 ± 0.00	0.28 ± 0.00
SyA	0.32 ± 0.01	0.27 ± 0.01	0.27 ± 0.00	0.29 ± 0.00	0.28 ± 0.00	0.27 ± 0.00	0.28 ± 0.00	0.31 ± 0.00	0.28 ± 0.00	0.28 ± 0.00
2,4,6-THBA	0.17 ± 0.00	0.17 ± 0.00	0.17 ± 0.00	0.17 ± 0.00	0.17 ± 0.00	0.17 ± 0.00	0.17 ± 0.00	0.17 ± 0.00	0.00 ± 0.00	0.18 ± 0.00
CouA	0.25 ± 0.01	0.21 ± 0.01	0.30 ± 0.00	0.24 ± 0.02	0.24 ± 0.00	0.20 ± 0.00	0.24 ± 0.00	0.26 ± 0.00	0.23 ± 0.00	0.27 ± 0.01
FA	0.31 ± 0.00	0.27 ± 0.00	0.29 ± 0.00	0.30 ± 0.00	0.30 ± 0.00	0.29 ± 0.00	0.29 ± 0.00	0.31 ± 0.00	0.29 ± 0.00	0.29 ± 0.00
BA	0.28 ± 0.02	0.32 ± 0.03	0.26 ± 0.00	0.31 ± 0.00	0.26 ± 0.00	0.25 ± 0.00	0.29 ± 0.00	0.24 ± 0.00	0.23 ± 0.00	0.26 ± 0.00
t-CiA	0.29 ± 0.01	0.19 ± 0.00	0.19 ± 0.00	0.27 ± 0.02	0.24 ± 0.00	0.19 ± 0.00	0.24 ± 0.00	0.25 ± 0.00	0.23 ± 0.00	0.24 ± 0.00
2-HBA	0.16 ± 0.00	0.21 ± 0.00	0.21 ± 0.00	0.18 ± 0.00	0.18 ± 0.00	0.21 ± 0.00	0.21 ± 0.00	0.17 ± 0.00	0.21 ± 0.00	0.21 ± 0.00
di-CQA	0.25 ± 0.00	0.22 ± 0.00	0.22 ± 0.00	0.24 ± 0.00	0.25 ± 0.00	0.22 ± 0.00	0.23 ± 0.00	0.23 ± 0.00	0.23 ± 0.00	0.22 ± 0.00
RoA	0.17 ± 0.00	0.17 ± 0.00	0.17 ± 0.00	0.17 ± 0.00	0.17 ± 0.00	0.17 ± 0.00	0.17 ± 0.00	0.17 ± 0.00	0.17 ± 0.00	0.17 ± 0.00
Flavonoids										
PB1	0.00 ± 0.00	1.19 ± 0.09	0.36 ± 0.00	0.53 ± 0.01	0.24 ± 0.00	0.78 ± 0.03	0.33 ± 0.00	0.27 ± 0.00	0.90 ± 0.03	0.29 ± 0.00
C	0.19 ± 0.00	3.12 ± 0.02	0.57 ± 0.02	1.54 ± 0.06	0.27 ± 0.00	2.02 ± 0.09	0.42 ± 0.01	0.36 ± 0.00	2.24 ± 0.09	0.24 ± 0.00
*epi*-C	0.19 ± 0.00	0.00 ± 0.00	0.19 ± 0.00	0.19 ± 0.00	0.19 ± 0.00	0.00 ± 0.00	0.19 ± 0.00	0.19 ± 0.00	0.19 ± 0.00	0.19 ± 0.00
R	0.20 ± 0.00	0.41 ± 0.00	0.61 ± 0.00	0.27 ± 0.01	0.45 ± 0.01	0.59 ± 0.02	0.40 ± 0.01	0.28 ± 0.00	0.38 ± 0.01	0.41 ± 0.01
Q-glur	0.19 ± 0.00	0.60 ± 0.01	0.54 ± 0.02	0.35 ± 0.00	0.43 ± 0.01	0.66 ± 0.02	0.43 ± 0.01	0.29 ± 0.00	0.49 ± 0.01	0.46 ± 0.01
Q-glu	0.27 ± 0.00	1.32 ± 0.05	0.62 ± 0.02	0.86 ± 0.03	0.46 ± 0.01	0.90 ± 0.03	0.68 ± 0.02	0.45 ± 0.01	0.88 ± 0.03	0.55 ± 0.01
Ta	0.17 ± 0.00	0.12 ± 0.00	0.00 ± 0.00	0.14 ± 0.00	0.12 ± 0.00	0.12 ± 0.00	0.12 ± 0.00	0.16 ± 0.00	0.13 ± 0.00	0.12 ± 0.00
K-glu	0.44 ± 0.00	3.25 ± 0.14	1.57 ± 0.05	1.10 ± 0.05	1.05 ± 0.03	2.52 ± 0.10	1.45 ± 0.05	0.63 ± 0.01	2.08 ± 0.08	1.48 ± 0.05
Hydrolyzable tannins										
DG-HHDP-Hex	0.00 ± 0.00	0.17 ± 0.03	0.04 ± 0.00	0.03 ± 0.00	0.03 ± 0.00	0.09 ± 0.00	0.05 ± 0.00	traces	0.07 ± 0.00	traces
TG-Hex	traces	0.05 ± 0.00	0.01 ± 0.00	0.01 ± 0.00	0.11 ± 0.00	0.17 ± 0.00	0.04 ± 0.00	0.01 ± 0.00	0.02 ± 0.00	0.04 ± 0.00
EA-Hex	0.00 ± 0.00	0.19 ± 0.10	0.11 ± 0.02	0.09 ± 0.01	0.19 ± 0.01	0.21 ± 0.01	0.17 ± 0.00	0.00 ± 0.00	0.15 ± 0.00	0.00 ± 0.00
PG-Glu	0.00 ± 0.00	0.19 ± 0.01	0.06 ± 0.00	0.03 ± 0.00	traces	0.12 ± 0.00	0.03 ± 0.00	0.00 ± 0.00	0.08 ± 0.00	0.02 ± 0.00
EA-Xyl	0.10 ± 0.02	2.10 ± 0.17	1.77 ± 0.33	0.96 ± 0.37	3.28 ± 0.15	4.08 ± 0.19	2.39 ± 0.11	1.11 ± 0.05	1.84 ± 0.08	2.94 ± 0.19
EA-Rham	0.01 ± 0.00	0.22 ± 0.01	0.09 ± 0.00	0.18 ± 0.01	0.18 ± 0.00	0.24 ± 0.01	0.24 ± 0.00	0.11 ± 0.00	0.13 ± 0.00	0.19 ± 0.00
EAME	0.00 ± 0.00	1.64 ± 0.09	0.02 ± 0.00	0.84 ± 0.14	0.02 ± 0.00	0.71 ± 0.03	0.58 ± 0.03	0.26 ± 0.01	1.22 ± 0.06	0.28 ± 0.01
EA-Hex iso	0.31 ± 0.02	28.34 ± 0.67	6.44 ± 1.84	18.05 ± 0.75	8.98 ± 0.42	22.92 ± 1.09	12.49 ± 0.59	5.25 ± 0.25	19.24 ± 0.91	10.07 ± 0.48
VeA	0.00 ± 0.00	0.02 ± 0.00	traces	traces	0.00 ± 0.00	0.01 ± 0.00	0.00 ± 0.00	0.01 ± 0.00	0.03 ± 0.00	0.01 ± 0.00
EA-Rham iso	0.99 ± 0.09	1.79 ± 0.05	0.87 ± 0.11	2.06 ± 0.95	2.26 ± 0.10	1.76 ± 0.08	1.45 ± 0.07	2.26 ± 0.11	1.34 ± 0.06	1.24 ± 0.06
EADME	2.17 ± 0.38	0.96 ± 0.03	1.01 ± 0.08	1.58 ± 0.50	2.27 ± 0.11	1.21 ± 0.05	1.36 ± 0.06	1.99 ± 0.09	1.23 ± 0.05	1.35 ± 0.06

Results are presented as mean ± standard deviation.

**Table 4 molecules-28-05167-t004:** Antioxidant activity of infusions prepared with mixtures of herbal sources.

Mixture	ORAC *	FRAP *	ABTS *	DPPH/Breaking-Chain **
*Y*	58.82 ± 7.16 ^d^	172.21 ± 0.93 ^b^	31.39 ± 2.22 ^e^	0.79 ± 0.04 ^g^
*Qs*	133.06 ± 13.72 ^a^	186.23 ± 1.31 ^a^	273.41 ± 3.15 ^a^	40.76 ± 1.72 ^a^
*Qe*	67.44 ± 5.72 ^c,d^	168.43 ± 0.93 ^c^	235.32 ± 22.21 ^b^	6.24 ± 0.27 ^c^
*Y-Qs*	57.40 ± 5.62 ^d^	174.49 ± 0.93 ^b^	204.21 ± 8.31 ^b^	5.56 ± 0.24 ^d^
*Y-Qe*	47.82 ± 0.79 ^e^	167.67 ± 0.93 ^c^	99.61 ± 6.28 ^d^	1.82 ± 0.08 ^g^
*Qs-Qe*	117.02 ± 9.34 ^a,b^	174.41 ± 1.61 ^b^	158.02 ± 3.15 ^c^	4.42 ± 0.19 ^e^
*Y-Qs-Qe*	60.36 ± 2.88 ^d^	176.17 ± 1.86 ^b^	103.24 ± 15.55 ^d^	1.34 ± 0.06 ^g^
*4Y-Qs-Qe*	53.42 ± 4.39 ^d,e^	170.52 ± 2.27 ^c^	118.62 ± 15.55 ^d^	2.38 ± 0.11 ^f^
*Y-4Qs-Qe*	98.29 ± 6.03 ^b^	176.06 ± 2.46 ^b^	148.33 ± 6.28 ^c,d^	9.32 ± 0.39 ^b^
*Y-Qs-4Qe*	76.28 ± 4.26 ^c^	169.86 ± 1.61 ^c^	89.51 ± 6.66 ^d^	1.24 ± 0.05 ^g^

Results are presented as mean ± standard deviation for (Tukey, *p* < 0.05; *n = 3*), different letters in each column indicate significant statistical difference. * expressed in µM ET/mL; ** expressed in −O.D.^−3^ min^−1^ mg^−1^.

## Data Availability

Data will be available upon request.

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
