# Peer review of "Analysis of Antioxidant Constituents of Filtering Infusions from Oak (Quercus sideroxyla Bonpl. and Quercus eduardii Trel.) and Yerbaniz (Tagetes lucida (Sweet) Voss) as Monoamine Oxidase Inhibitors"

_molecules, 2023, doi:10.3390/molecules28135167_

Round 1

Reviewer 1 Report

The work presented in this paper appears to be of good quality and could be of interest to the scientific community.
The manuscript is very interesting as it explores unconventional sources to identify the potential for MAO-A inhibition. Also, the authors report an oak source with a phenolic quality that is related to the capacity of this enzyme, which may lead to future commercialization as this source is reported by the authors to have a history of consumption by the population.

Derived from the analysis performed by the researchers, was report that some hydrolyzable tannins are able to inhibit MAO-A, which is an important contribution.

Finally, I suggest that the material considered as complementary can be  integrated into the document.The work presented in this paper appears to be of good quality and could be of interest to the scientific community.

Author Response

In the attached word document, you will find the attention of the comments made to the manuscript.

Reviewer 2 Report

This paper aimed to investigate the antioxidant constituents of filtering infusions from some oak species and yerbaniz as monoamine oxidase inhibitors. The manuscript is interesting and well-organized. The author’s work on discussing achieved results is appreciated. The title is clear and adequate to the article’s content. The conclusions or summary are supported by the content. I think that revisions are necessary to improve the presentation’s clarity and make convincing scientific arguments.

I have some recommendations for authors:

- I suggest using the completed name of the plant species - Quercus sideroxyla Bonpl., Quercus eduardi Trel., Tagetes lucida (Sweet) Voss

- Please explain the abbreviation MAO-A from the abstract.

- I suggest that the keywords chosen are not from the title.

- Please highlight the degree of novelty and originality of the work.

- Please include the voucher number for plant material.

- The methods are original? If not, please include citations.

- Check the bibliography to be written according to the requirements of the journal.

- Include in the conclusions potential research directions. What are the future applications? What are the next research directions?

Author Response

(The authors gave the same response as above.)

Reviewer 3 Report

Review Report:

Analysis of antioxidant constituents of filtering infusions from oak (Quercus sideroxyla and Quercus eduardii) and yerbaniz (Tagetes lucida) as Monoamine Oxidase Inhibitors

This paper comprehensively presents to evaluate in vitro and in silico the effect of antioxidant constituents of filtering infusions from yerbaniz (Tagetes lucida) and oak (Q. sideroxyla and Q. eduardii) as monoamine oxidase inhibitors. This research contemplates the study of filtering infusions with yerbaniz (Tagetes lucida), oak leaves (Quercus sideroxyla and Q. eduardii) and their mixtures, chosen based on their composition of hydroxycinnamic acids and flavonols, chemical groups recently related to anxiolytic and antidepressant effects In this original research manuscripts, the work has been reported scientifically sound experiments and provides a substantial amount of new information.

1.      The title of your manuscript is concise, specific and relevant.

2.      The paper is well written and presents original research. However, authors have not followed the journal’s specific author guidelines. Please revise the paper accordingly.

3.      Please correct “in text citations” throughout the manuscript. These are not according to journal format.

4.      The technical language of the manuscript is good. The few grammatically errors should be checked but overall language and style of the paper should be revisited.

5.      Abstract can be modified, and authors are required to add the quantitative findings in the abstract (2-3 lines).

6.      The abstract should be an objective representation of the article: it must not contain results which are not presented and substantiated in the main text and should not exaggerate the main conclusions.

7.      Please write concluding remarks and future recommendation at the end of abstract

8.      SI units (International System of Units) should be used. Imperial, US customary and other units should be converted to SI units whenever possible.

9.      Alphabetically arrange the keywords

10.  Please rewrite sentence: Depression and anxiety are problems that have affected all health systems world-wide because these are diseases that have positioned themselves among the most frequent.

11.  Give a little background justification of study, as a review of literature (2-3 lines) in introduction section before aims and objectives.

12.  Material and methods are described with sufficient detail to allow others to replicate and build on published results. New methods and protocols have been described in detail while well-established methods can be briefly described and appropriately cited.

13.   There are few or no references in whole material and methods section. Please add.

14.  Provide a concise and precise description of the experimental results, their interpretation as well as the experimental conclusions that can be drawn.

15.  Improve the manuscript’s discussion thoroughly. Authors should discuss the results and how they can be interpreted in perspective of previous studies and of the working hypotheses. The findings and their implications should be discussed in the broadest context possible and limitations of the work highlighted. Future research directions may also be mentioned.

16.  Please include a critical discussion on the key findings of the study.

17.  Conclusion is too brief, insufficient to conclude the results of study. Rewrite the quantitative conclusion please

18.  Future perspectives should be written as last paragraph in conclusion section. Do not merge future outlooks in original findings.

19.  References must be numbered in order of appearance in the text (including table captions and figure legends) and listed individually at the end of the manuscript.

20.  We recommend preparing the references with a bibliography software, EndNote, to avoid typing mistakes and duplicated references.

21.  In the text, reference numbers should be placed in square brackets [ ], and placed before the punctuation; for example [1], [1–3] or [1,3]. For embedded citations in the text with pagination, use both parentheses and brackets to indicate the reference number and page numbers. Accepted for publication after minor revision. 

Moderate English improvement is needed. 

Author Response

(The authors gave the same response as above.)

Round 2

Reviewer 2 Report

The authors responded adequately to all my comments.